# MULTITURNCLEANUP: A Benchmark for Multi-Turn Spoken Conversational Transcript Cleanup

**Hua Shen**♥♦[*]   **Vicky Zayats**♦   **Johann C. Rocholl**♦   **Daniel D. Walker**♦   **Dirk Padfield**♦
♥University of Michigan, ♦Google Research
huashen@umich.edu
{vzayats,jcrocholl,danwalkeriv,padfield}@google.com

## Abstract

Current disfluency detection models focus on individual utterances each from a single speaker. However, numerous discontinuity phenomena in spoken conversational transcripts occur across multiple turns, which can not be identified by disfluency detection models. This study addresses these phenomena by proposing an innovative Multi-Turn Cleanup task for spoken conversational transcripts and collecting a new dataset, MULTITURNCLEANUP[1]. We design a data labeling schema to collect the high-quality dataset and provide extensive data analysis. Furthermore, we leverage two modeling approaches for experimental evaluation as benchmarks for future research.

## 1 Introduction

Spontaneous spoken conversations contain interruptions such as filled pauses, self-repairs, etc. (Shriberg, 1994). These phenomena act as noise that hampers human readability (Adda-Decker et al., 2003) and the performance of downstream tasks such as question answering (Gupta et al., 2021) or machine translation (Hassan et al., 2014) on transcripts of human spoken conversations. State-of-the-art disfluency detection methods (Yang et al., 2020; Jamshid Lou and Johnson, 2020) identify and remove disfluencies in order to improve the readability of spoken conversational transcripts (Wang et al., 2022; Chen et al., 2022). For instance, Figure 1(a) shows that disfluency detection methods can remove self-repairs of single turns. However, these models focus on removing interruptions and errors that commonly occur within single-turn utterances and cannot handle discontinuities across multiple turns. For example, in Figure 1(b), speaker **B** is in the middle of a thought when Speaker **A** interrupts to signal that they are

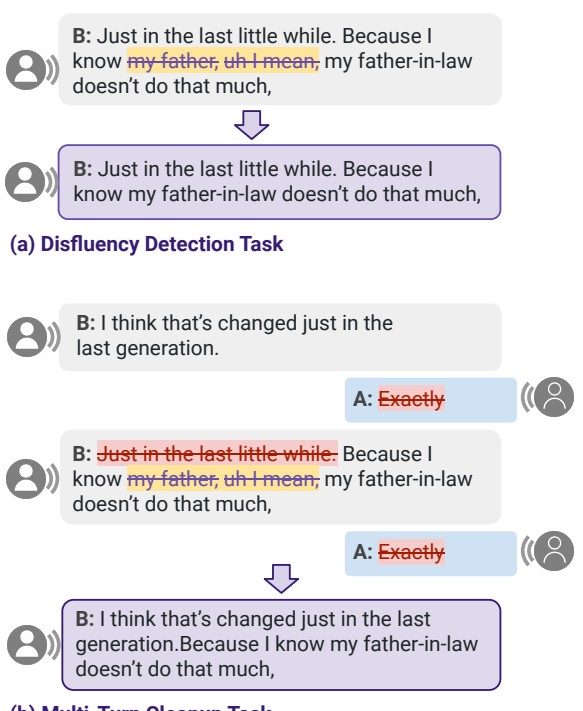

Figure 1: A comparison of (a) existing Disfluency Detection Task (yellow highlights indicate disfluencies) with (b) the proposed Multi-Turn Cleanup task (red highlights indicate multi-turn cleanups) for spoken conversational transcripts.

following along ("**A**: Exactly"). **B** continues their train of thought ("**B**: Just in the last little while. Because...") by paraphrasing their own last sentence ("...just in the last generation."). The result is an exchange that is longer and more difficult to follow than necessary to understand what **B** is conveying.

This paper aims to "clean up" spoken conversation transcripts by detecting these types of multi-turn "discontinuities" inherent in spontaneous spoken conversations. Once detected, they can be removed to produce transcripts that look more like hand-written conversations conducted over text messaging, social media, or e-mail as shown in Figure 1(b). Given that this is a novel task, with no

---

[*]This work was done when the first author was a research intern at Google Research.

[1]We release the collected MULTITURNCLEANUP dataset at: https://github.com/huashen218/MultiTurnCleanup.git

| Category | Definition | Count (%) | Conversation Instance |
|---|---|---|---|
| Acknowledgment and Confirmation | Speakers show that they are listening to and agree with the other speakers | 24.3k (17%) | **A**: I guess both of us are very much aware of the equality. it seems like women are, just starting to get kind of equality in jobs and the home where husbands are starting to doing dishes, ~~or some~~ |
| Repetition and Paraphrase | Speakers may repeat or paraphrase their words during the conversation. | 30k (21%) | **B**: I think that's changed just in the last generation.
**A**: ~~Exactly.~~
**B**: ~~Just in the last little while.~~ Because I know my father-in-law doesn't do that much, |
| Think aloud | Speakers talk to themselves during thinking instead of talking to others. | 15.7k (11%) | **A**: ~~Exactly.~~
**B**: of dishes, taking care of kids, ~~or what else, you know,~~ that kind of stuff but my husband is wonderful. |
| Incomplete Sentences | Speakers may also say incomplete sentences due to interruption, changing topics, etc. | 47.2k (33%) | **A**: that's the way my husband is too. it doesn't bother him to do the dishes, ~~it doesn't bother him to do~~ the laundry verses, men from way back |
| Others | The remaining discontinuity categories. | 25.8k (18%) | , ~~there is that,~~ if you did that you were henpecked. |

Table 1: The linguistic taxonomy (**Category**) and definition (**Definition**) of discontinuities in the MULTITURN-CLEANUP dataset for the Multi-Turn Cleanup task. We further provide the statistics of each category (**Count(%)**) in the dataset and a conversational instance (**Conversation Instance**), where **A** and **B** indicate two speakers.

pre-existing labeled data or benchmarks, we first define a taxonomy of non-disfluency discontinuities (see Figure 1). Then we collect a new dataset, MULTITURNCLEANUP, for the **Multi Turn** spoken conversational transcript **Cleanup** task, based on the Switchboard Corpus (Godfrey et al., 1992) and label according to the proposed taxonomy. Finally we develop two baseline models to detect these discontinuities which we evaluate as benchmarks for future Multi-Turn Cleanup task studies. Our data analysis suggests that the MULTITURNCLEANUP dataset is of high quality. We believe it will help to facilitate research in this under-investigated area.

## 2 Data Collection and Analysis

We propose an innovative Multi-Turn Cleanup task and collect a novel dataset for this task called MULTITURNCLEANUP[2]. This section presents the task definition, data collection process, and analysis.

### 2.1 Task Definition

Compared with the existing *disfluency detection* task, which aims to detect disfluencies (*e.g.,* self-repairs, repetitions, restarts, and filled pauses) that commonly occur within single-turn utterances (Rocholl et al., 2021; Chen et al., 2022), the **Multi-Turn Cleanup task** requires identifying discontinuities both within a single turn and across multiple turns in the multi-party spoken conversational

transcripts. To explicitly define the task and discontinuity taxonomy, we conducted an in-depth analysis of the Switchboard corpus[3] (Godfrey et al., 1992). Specifically, we randomly sampled a subset of Switchboard conversations, annotated the discontinuity spans other than existing disfluency types, and grouped the annotated discontinuities into five main categories. Note that we conducted the discontinuity annotation and category grouping process iteratively with all authors to reach the consensus. We demonstrate the finalized taxonomy of discontinuities in Table 1.

### 2.2 Data Preprocessing

We preprocessed the Switchboard corpus by automatically removing the single-turn disfluencies with pre-defined rules based on Treebank-3 (Mitchell et al., 1999)[4]. As a result, we could encourage our recruited humans annotators to concentrate on cleaning up multi-turn discontinuity phenomena based on the five categories in Table 1.

We split each conversation into multiple chunks, where each chunk composes one HIT (Human Intelligence Task) containing around 300 tokens. We further ensure that the successive chunks overlap around 50% of tokens, providing enough context for each conversation fragment. The resulting data

---

[2]The data collection cost was about $18,000, with payment as $0.40/HIT ($8/hour assuming 3min/HIT) and bonuses ($2 − $100/worker) for top raters.

[3]Switchboard corpus is a collection of five-minute human-human telephone conversations. We chose Switchboard for data construction because: 1) it has relatively large size; and 2) it contains ground-truth disfluency annotations that our study can build upon to annotate multi-turn cleanup labels.

[4]See Appendix A.2 for detailed preprocessing rules and data statistics.

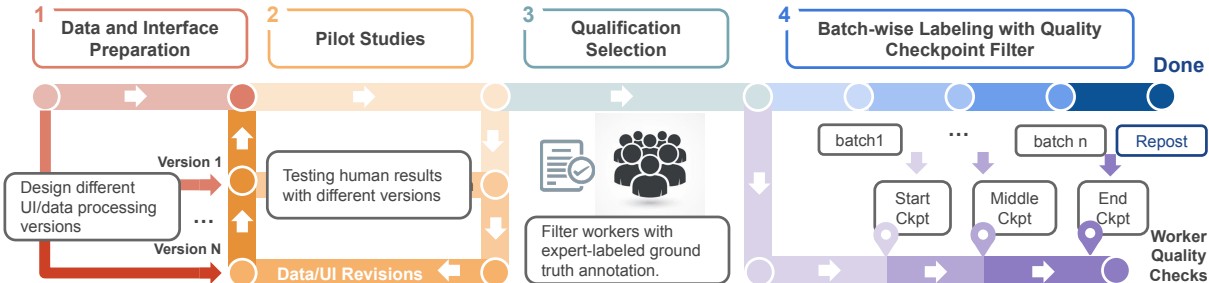

Figure 2: A four-step data labeling schema to collect MULTITURNCLEANUP dataset for the Multi-Turn Cleanup task on spoken conversational transcripts. This schema enabled efficient collection of the high-quality MULTITURN-CLEANUP dataset via MTurk platform, where annotators labeled cleanup marks and corresponding categories.

| Datasets | #Conv | #Turns | #Tokens | #Cleanup |
|----------|-------|--------|---------|----------|
| **Train** | 932 | 74k | 1M | 132k |
| **Dev** | 86 | 3.7k | 60k | 6.1k |
| **Test** | 64 | 2.9k | 43k | 5k |
| **Sum** | 1082 | 81k | 1.1M | 143k |

Table 2: Statistics of MULTITURNCLEANUP dataset.

| IRR | Experts | MTurk Workers | | | |
|-----|---------|------|-----|------|-----|
| | | **Train** | **Dev** | **Test** | **All** |
| **Fleiss' Kappa** | 0.596 | 0.560 | 0.592 | 0.557 | 0.561 |

Table 3: The averaged Fleiss' Kappa scores of all conversation turns in the MULTITURNCLEANUP dataset.

preprocessing statistics are shown in Table 6 in Appendix A.2.

## 2.3 Labeling Procedure

Given the preprocessed data, we then conducted the human annotation process based on a data labeling schema[5] shown in Figure 2.

**Preparation and qualification selection.** In steps 1 and 2, we prepared a suite of data preprocessing and user interface (UI) variations and conducted seven pilot studies to select the optimal task design. The final UI (see Appendix A.4) consists of: *i)* an introduction to the task, *ii)* an annotation example with highlighted discontinuities, and *iii)* the task workspace with affordances for annotation. In step 3, we recruited a set of qualified MTurk workers using a "Qualification HIT". We compared all 580 workers' submissions with the ground truth (authors' consensus) and select the 222 workers (38.3%) with an $F1 \geq 0.3$[6] to participate in step 4.

**Large-scale data labeling.** Controlling annotation quality for large-scale data labeling is challenging in MTurk (Daniel et al., 2018). To address this, we employed a "batch-wise labeling with quality checkpoint filter" (Bragg and Weld, 2016). Specifically, we split the dataset into small batches and posted them with "Quality Checkpoint HITs" (QCH) mixed in. Overall, we posted 22

batches including 7277 HITs and 11 QCH in total. We leverage these checkpoint HITs to exclude unqualified workers ($F1 \leq 0.3$).

**Annotation filtering and aggregation.** After finishing the final batch, we collected all annotated batches and excluded 72 unqualified workers with all their HITs. Then we reposted 26% of the assignments where the conversations had less than two annotations to the remaining qualified workers. Finally, we aggregated the annotations for each turn by only keeping the best worker's (highest F1-score) labels to compose the MULTITURNCLEANUP dataset. The average F1 for raters of labeled turns in MULTITURNCLEANUP is 0.57. We summarize the per-category statistics in Table 1 and the statistics of MULTITURNCLEANUP in Table 2[7].

## 2.4 Validating Human Annotation Accuracy

During the whole data labeling process, we consistently assessed the human annotation accuracy and filtered out unqualified workers to control the data quality. We visualize the annotation quality in terms of the distribution of workers' F1 scores (see Figure 3(A)(B)-left), as well as the correlation between each worker's F1 score and their finished HIT counts and the average elapsed time per HIT (see Figure 3(A)(B)-right). These figures show how removing unqualified annotations with checkpoints can effectively control quality during the annota-

---

[5]Steps 3 and 4 lasted about one month. More annotation quality control details are available in Appendix A.3.

[6]The 0.3 threshold is reasonable due to task subjectivity.

[7]We leave out sw4[2-4]* subgroups in Switchboard as they are less commonly used, resulting in 1082 total conversations.

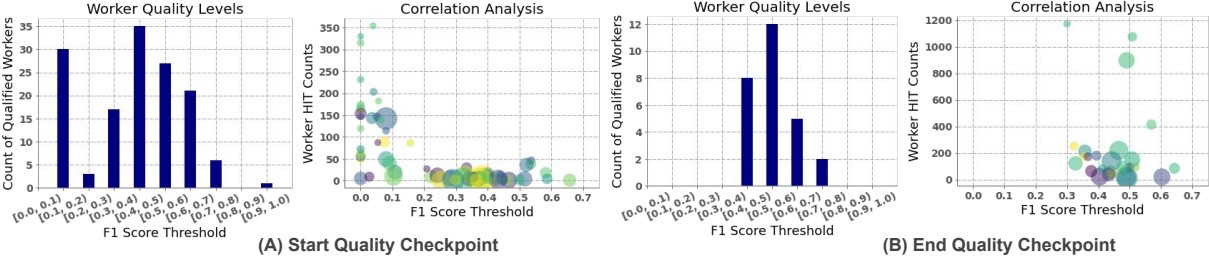

**(A) Start Quality Checkpoint**    **(B) End Quality Checkpoint**

Figure 3: The comparison of worker quality performance between (A) start checkpoint and (B) end checkpoints. For each one, we plot participated workers' F1 score distribution (left) and the correlation between each worker's F1 score and finished HIT count (right), the circle size means each worker's averaged elapsed time to finish a HIT.

| Sub-tasks | Model | F1 | R | P |
|---|---|---|---|---|
| DISFLUENCY | STD | 89.8 | 88.3 | 91.2 |
| MULTITURN CLEANUP | BASELINE | 15.5 | 8.77 | 65.8 |
| | MTD | 56.8 | 55.4 | 58.3 |

Table 4: Model performance on the two sub-tasks, including detecting single-turn disfluencies with DISFLUENCY dataset and multi-turn discontinuities with the proposed MULTITURNCLEANUP dataset.

| | Model | F1 | R | P |
|---|---|---|---|---|
| **Multi-Turn Cleanup Task** | BASELINE | 58.2 | 42.5 | 92.3 |
| | Two-Stage | 68.2 | 64.6 | 72.3 |
| | Combined | 74.9 | 72.9 | 76.9 |

Table 5: Modle performance on the overall Multi-Turn Cleanup task with the UNIONDISCONTINUITY dataset, where the Combined Model achieves the best F1 score.

tion process. Particularly, we observe that at the start (A), even after passed our initial "Qualification HIT" in step 3, 23% of workers perform at F1 < 0.3 but complete over 80% of all assignments, leaving only a limited amount of data for more competent workers to label. By continually excluding unqualified workers with F1 < 0.3, all remaining workers have F1 ≥ 0.3 by the final batch (B).

## 2.5 Turn-based Inter-Rater Reliability

We compute Inter-Rater Reliability using Fleiss' Kappa Agreement (Fleiss and Cohen, 1973) for each annotated turn and average all turns' scores. Table 3 shows that the workers' Fleiss' Kappa scores are comparable to those of the authors.

## 3 Multi-turn Cleanup Models

Given the collected MULTITURNCLEANUP dataset, we leverage two different BERT-based modeling approaches, including a two-stage model and a combined model, for the Multi-Turn Cleanup task to remove both single-turn disfluencies and multi-turn discontinuities.

## 3.1 The Two-Stage Model

The two-stage model is composed of a Single-Turn Detector (STD) to remove the traditional single-turn disfluencies and a successive Multi-Turn Detector (MTD) to remove the discontinuities occurring across multiple turns. We employ the

BERT-based modeling, presented in Rocholl et al. (2021), for both STD and MTD stages. Particularly, we fine-tune the STD based on the traditional single-turn disfluency dataset (Godfrey et al., 1992), whereas the MTD is fine-tuned based on our collected MULTITURNCLEANUP dataset. We concatenate STD and MTD successively into the pipeline of the two-stage model, so that both the single-turn disfluencies and multi-turn discontinuities in the raw conversational transcript can be removed with one pass.

## 3.2 The Combined Model

We design the combined model, using only one BERT-based detector (Rocholl et al., 2021), to simultaneously remove both single-turn disfluencies and multi-turn discontinuities. To this end, we create a UNIONDISCONTINUITY dataset, which combines both the single-turn disfluency and multi-turn discontinuities labels in Godfrey et al. (1992) and MULTITURNCLEANUP datasets, respectively. Then we achieve the combined model by fine-tuning the detector with this UNIONDISCONTINUITY dataset.

## 4 Experiments

## 4.1 Experimental Setup

**The Two-Stage Model**. The STD and MTD are separately trained. We train the STD with the existing disfluency dataset, where the input is a single sentence (*i.e.,* slash unit), with a maximum sequence length of 64. In comparison, we train the

MTD with MULTITURNCLEANUP dataset, where the input consists of multiple slash-units (demarcated with [SEP] token between turns) with a maximum sequence length of 512. We feed full transcripts to the MTD in chunks with an overlap of 50% for prediction context. Then we predict discontinuities where either of the overlapping predictions for a given token was positive. During inference, the stage-2 MTD module loads the outputs from the stage-1 STD module, removes all of the tokens classified as disfluencies, and uses this redacted texts as its own input.

**The Combined Model**. We train the combined model with the UNIONDISCONTINUITY dataset using the same training settings of the aforementioned MTD module. During inference, we predict both single-turn and multi-turn discontinuities, as nondistinctive labels, simultaneously.

**Baseline**. We employ the state-of-the-art BERT based disfluency detection model (Rocholl et al., 2021) trained with the widely used disfluency dataset (Godfrey et al., 1992) as the BASELINE.

**Deployment**. We train the models on Google's AutoML platform, where it selects the optimal training settings as: Adam optimizer with learning rate as $1e-5$, batch size of 8, and 1 epoch.

### 4.2 Evaluation Metrics

We evaluate all models' performance with per-token Precision (**P**), Recall (**R**), and F1 score (**F1**) on predicting if the token should be cleaned as single-turn disfluencies (STD of two-stage model), or multi-turn discontinuities (MTD of two-stage model), or their mixtures (the combined model).

### 4.3 Results

**Evaluation on two sub-tasks**. The Multi-Turn Cleanup task inherently involves two different sub-tasks, including the single-turn disfluency detection (*i.e.,* with DISFLUENCY dataset) and multi-turn discontinuity detection (*i.e.,* with our collected MULTITURNCLEANUP dataset), we first validate that the presented models can achieve state-of-the-art performance on the two sub-tasks (*i.e.,* with the two different datasets), respectively.

Particularly, Table 4 illustrates the performance of BASELINE and presented models on the two datasets. The STD module achieves cutting-edge performance (Chen et al., 2022) to detect single-turn disfluencies. Also, the MTD module outperforms the BASELINE on detecting multi-turn discontinuities with our proposed MULTITURNCLEANUP

dataset. The significant disparity between MTD and BASELINE methods (*e.g.,* 56.8 vs. 15.5 in F1) also indicate the difficulty of detecting multi-turn discontinuities in MULTITURNCLEANUP dataset.

**Evaluation on removing all discontinuities**. Furthermore, we evaluate the overall model performance on jointly detecting the single-turn disfluencies and multi-turn discontinuities with one pass based on the UNIONDISCONTINUITY dataset. As shown in Table 5, we observe that both the proposed Two-Stage Model and Combined Model can outperform BASELINE method. In addition, the Combined Model achieves a 6.7 higher F1 score than Two-Stage Model on the Multi-Turn Cleanup task.

## 5 Related Work

Recent disfluency detection studies develop BERT-based models (Bach and Huang, 2019; Rocholl et al., 2021; Rohanian and Hough, 2021) and show significant improvement over LSTM-based models (Zayats et al., 2016; Wang et al., 2016; Hough and Schlangen, 2017) in disfluency detection tasks. Prior studies also show the importance of data augmentation methods, by leveraging extra transcript sources to improve disfluency detection performance (Jamshid Lou and Johnson, 2017, 2020). While most of the research has been focused on improving single-turn disfluency detection accuracy, little exploration has been done in detecting multi-turn transcript discontinuities.

Obtaining reliable annotated datasets via crowd-sourcing is challenging and expensive (Alonso et al., 2014; Wong et al., 2022; Northcutt et al., 2021). To collect qualified dataset for multi-turn cleanup task, this work designs a data labeling schema which efficiently collects qualified dataset via the MTurk.

## 6 Limitation and Conclusion

We are aware that, in some specific scenarios, it might be undesirable to remove some multi-turn discontinuities because they convey social meaning in human interactions (*e.g.,* engagement). We address this issue by providing category labels. As a result, future research can flexibly select subsets of the MULTITURNCLEANUP labels to train the model and clean up multi-turn discontinuities.

This study defines an innovative Multi-Turn Cleanup task and collects a high-quality dataset for this task, named MULTITURNCLEANUP, using our presented data labeling schema. We further leverage two modeling approaches for experimental evaluation as the benchmarks for future research.

## Acknowledgements

We thank Noah B. Murad for his help in conducting the human evaluation experiments. We also thank Shyam Upadhyay, Daniel J. Liebling, Tiffany Knearem, Kenton Lee, and other Google colleagues for providing their constructive feedback on this study. We thank the Amazon MTurk workers for their excellent annotations. We thank the reviewers for their thoughtful comments.

## Ethics Statement

The collected MULTITURNCLEANUP dataset is built upon the published Switchboard Corpus (Godfrey et al., 1992). The dataset is sufficiently anonymized, so it is impossible to identify individuals. In addition, we protect privacy during the data collection process through the MTurk platform. The posted dataset does not list any identifying information of MTurk workers. Also, the data collection process does not access any demographic or confidential information (*e.g.,* identification, gender, race, etc.) from the MTurk workers. In general, the dataset can be safely used with low risk in research and application fields for cleaning up spoken conversations and speech transcripts.

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

# A  Appendix

## A.1  Definitions of Disfluency Detection and Annotations

**Disfluency Definition.** When humans speak, our language is peppered with interruptions and errors known as disfluencies. Formally, disfluencies are irregularities that are an integral part of spontaneous speech and include *self-repairs*, *repetitions*, *restarts*, and *filled pauses* (Schegloff et al., 1977).

**Disfluency Annotations.** Following Shriberg et al. (1997), the disfluency annotation includes:

- the ***reparandum***: the material that the speaker intends to delete.

- the ***interruption point***: denoted as (+).

- optional ***interregnum***: enclosed in curly brackets, which include *filled pauses* and *discourse markers*, such as "uh", "um", "you know", "I mean", etc.

- optional ***repair***: the material that semantically replaces the reparandum.

Some examples of disfluency annotation include:

- `[ it's + { uh } it's ] almost ...`

- `[ was it, + { I mean, } did you ] put...`

- `[ I just + I ] enjoy working`

- `[ By + ] it was attached to ...`

## A.2  Dataset Preprocessing Details.

| Datasets | #Conv | #HITs | #Turns | #Tokens |
|---|---|---|---|---|
| **Train** | 932 | 6579 | 142,135 | 1,928,169 |
| **Dev** | 86 | 396 | 7,250 | 115,756 |
| **Test** | 64 | 302 | 5,888 | 85,321 |
| **Sum** | **1082** | **7277** | **155,273** | **2,129,246** |

Table 6: Statistics of data preprocessing results.

We preprocess the Switchboard corpus (Godfrey et al., 1992) with the human-annotated disfluencies based on Treebank-3 (Mitchell et al., 1999).

Based on the human annotations, we first cleaned the sentences by removing the non-speech events and words/phrases with the markers including:

- Prosodic markup, like # and /.

- Other language markup, like <English bike>.

- Noise markers, like «laughter» or <Throat_clearing>

- Noise markers with curly brackets, like {breathing} or {gasp} or {lipsmack} or {again, imitates the sound of whales}

- Double parentheses from uncertainty markers, like ((yesterday))

- Plus signs surrounding words, like +sight-seeing+.

- Context markers, like (laughter) or (RE-CESS).

- Punctuations, like %, **, &, />, +>, <]>, (), ((, )), [[, ]].

In the next stage, we conservatively remove the following types of disfluencies recursively:

- Reparandum – the material that the speakers intends to delete.

- Interregnum – including types of:

  - Discourse markers (*i.e.,* marked with {**D**...} like *you know*, *well*, *so*, *like*, etc.).
  - Explicit editing term (*i.e.,* marked with {**E**...} like *I mean*, *sorry*, *excuse me*, etc).
  - Filler words (*i.e.,* marked with {**F**...} like *uh*, *um*, *huh*, *oh*, etc.

By removing all the markers and disfluencies described above, we present the remaining contents of Switchboard corpus to MTurk workers for labeling the MULTITURNCLEANUP dataset for Multi-Turn Cleanup task.

### A.3 Key notes of data labeling.

We notice a list of key points that are imperative to ensure the data quality and avoid spammers in MTurk platform.

- *Keep task simple and instructions clear.* This helps workers to better understand the task, reduce their cognitive load and focus on annotating qualified outputs.

- *Select workers and check quality constantly.* Repeatedly inspecting worker quality can significantly reduce spammers and improve data quality, as more details validated in Sec 2.4.

- *Communicate via emails.* Notifying new post or notes to workers helps improve return rate significantly. Also, their feedback may be valuable to improve the data labeling.

### A.4 User Interface Demo

We demonstrate the finalized User Interface (UI) design of the HIT for the large-scale datasets in Figure 4.

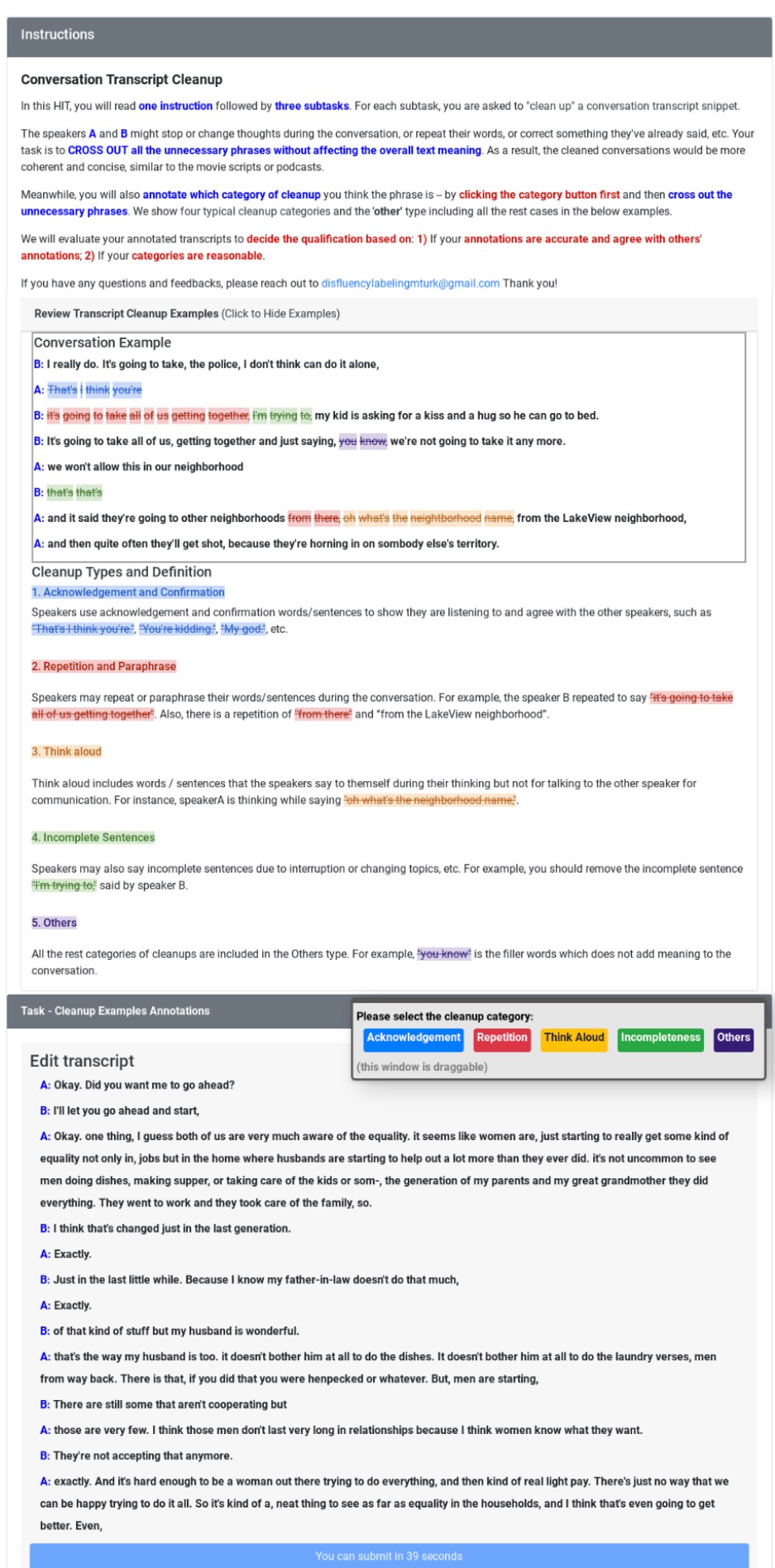

Figure 4: An example of the User Interface for the MTurk worker annotation. The "Review Transcript Cleanup Example" section is foldable by clicking to hide and show the example.