# OpenReview forum: "MultiTurnCleanup: A Benchmark for Multi-Turn Spoken Conversational Transcript Cleanup"
_EMNLP/2023/Conference — EMNLP 2023 Main_

### Official Review · Reviewer_Hvuk · 2023-08-04

**Soundness:** 3

**Excitement:**

3: Ambivalent: It has merits (e.g., it reports state-of-the-art results, the idea is nice), but there are key weaknesses (e.g., it describes incremental work), and it can significantly benefit from another round of revision. However, I won't object to accepting it if my co-reviewers champion it.

**Missing References:**

Na

**Paper Topic And Main Contributions:**

This paper proposes a multi-turn cleanup dataset to remove the numerous discontinuity during the spoken dialogues. Multi-turn cleanup is sort of disfluency detection on the dialogue level instead of utterance level. The paper carefully designs a schema to collect the high-quality data. This dataset might be quite helpful to the study of disfluency detection in the dialogue domain

**Questions For The Authors:**

na

**Reasons To Accept:**

1 This paper is well written and easy to follow
2 The labelling schema is well designed and the problem of disfluency in the spoken dialogues is an interesting topic to explore.
3 The authors build a fairly solid baseline for other works to follow-up

**Reasons To Reject:**

1 Lack of analysis by using the latest LLMs
Maybe this problem can be very-well addressed by the LLMs, the authors should post a result by using LLM. I give the conversation instance in table1 to gpt 4, and show it a few examples of discontinuity, and it did quite a good job.

**Reproducibility:**

3: Could reproduce the results with some difficulty. The settings of parameters are underspecified or subjectively determined; the training/evaluation data are not widely available.

**Reviewer Confidence:**

3: Pretty sure, but there's a chance I missed something. Although I have a good feel for this area in general, I did not carefully check the paper's details, e.g., the math, experimental design, or novelty.

---

> ### Author Rebuttal · Authors · 2023-08-27
>
> We thank the reviewer for your constructive feedback and suggestions.
>
> We only leveraged Bert-based models mainly due to two reasons:
>
> 1. The Bert-based models showed state-of-the-art performance in the latest disfluency detection studies, which is the most common approach [1,2]. As our work’s main contribution is to create a dataset for multi-turn discontinuity cleanup and to build the basic common models for a benchmark, we focused on checking the most widely-used models’ results instead of pursuing the highest performance.
>
> 2. The GPT4 had not been released yet when we started to design and conduct our large-scale modeling and evaluation experiments.
>
> We thank the reviewer for sharing their insights and results of giving the conversation instance to GPT4. This interesting and promising finding also potentially indicates the validity of our proposed novel dataset and task.
>
> Currently, we believe that using Bert-based models is sufficient to serve our goal: illustrating our novel task’s solvability and difficulty compared with classical approaches. However, we are also interested in exploring more modeling innovations with LLMs on our contributed data and task in future work. We will describe the reviewer’s finding and encourage this promising direction in our revised paper version too.
>
> References:
>
> [1] Rocholl, Johann C., et al. "Disfluency detection with unlabeled data and small BERT models." arXiv preprint arXiv:2104.10769 (2021).
>
> [2] Morteza Rohanian and Julian Hough. 2021. Best of Both Worlds: Making High Accuracy Non-incremental Transformer-based Disfluency Detection Incremental. In Proceedings of the 59th Annual Meeting of the Association for Computational Linguistics and the 11th International Joint Conference on Natural Language Processing (ACL-IJCNLP), pages 3693–3703, Online. Association for Computational Linguistics.

---

### Official Review · Reviewer_H2MH · 2023-08-05

**Soundness:** 4

**Excitement:**

4: Strong: This paper deepens the understanding of some phenomenon or lowers the barriers to an existing research direction.

**Paper Topic And Main Contributions:**

The paper proposes to clean up spoken conversation transcripts by detecting multi-turn discontinuities found in spontaneous spoken conversations.

The authors first define a taxonomy of multi-turn discontinuities and then collect a dataset based on this taxonomy.

The authors then train 2 baseline models using this dataset and find that the dataset can be used to facilitate future research in building multi-turn disfluency detection systems.

**Reasons To Accept:**

1. Introduction of new dataset that will facilitate future research in multi-turn discontinuity detection, an important problem in my opinion.
2. Very detailed description of their data annotation process that can benefit future data collection experiments.

**Reasons To Reject:**

1. It would have been interesting if they had also included audios in their dataset since some recent work has shown that non-phonemic information like prosody can improve disfluency detection performance.

**Reproducibility:**

5: Could easily reproduce the results.

**Reviewer Confidence:**

4: Quite sure. I tried to check the important points carefully. It's unlikely, though conceivable, that I missed something that should affect my ratings.

---

> ### Author Rebuttal · Authors · 2023-08-24
>
> We thank the reviewer for your insightful comments and suggestions. Although audio data is beyond this short paper’s scope, adding non-phonemic information like prosody to improve disfluency detection performance seems to be a promising direction, and we would like to plan this exploration in our future work.

---

### Official Review · Reviewer_chnj · 2023-08-11

**Soundness:** 4

**Ethical Concerns:**

Yes

**Excitement:**

3: Ambivalent: It has merits (e.g., it reports state-of-the-art results, the idea is nice), but there are key weaknesses (e.g., it describes incremental work), and it can significantly benefit from another round of revision. However, I won't object to accepting it if my co-reviewers champion it.

**Justification For Ethical Concerns:**

While the authors state how much the annotators (turkers) were paid, it is open, whether they were paid adequately, if they were treated fairly etc.

**Missing References:**

References to work on Disfluencies related to the ISCI and TRAINS corpus, i.e. https://aclanthology.org/N13-1083.pdf might be a good starting point.

**Paper Topic And Main Contributions:**

The paper "MultiTurnCleanup: A Benchmark for Multi-Turn Spoken Conversational Transcript Cleanup" presents a workflow to tackle disfluencies not only in single utterances/utterances by the same speaker, but also disfluencies spread across multiple utterances/several speakers. To that end, the paper presents an annotation scheme, manual annotations and an automatic detection model for finding and removing all disfluencies. The experiments are based on the Switchboard corpus, which has been labelled according to the presented annotation scheme.


**Questions For The Authors:**

The work is conducted on the Switchboard corpus. Is there a reason, the ICSI corpus has not been used? In the past, it has been studied extensively for disfluencies. Similar to the TRAINS corpus, where also a range of scientific papers have been published on this topic.

What is the goal of removing disfluencies in such a way? Is it to aid summarization? To improve some other down-stream tasks?

Why was the data annotated with the specific annotation scheme? A lot of the phenomena covered in the annotations, could also be handled using Dialog Acts and more specifically, Backchannels, which some of these phenomena are.

The paper reports the IAA based on Fleiss Kappa for the annotation. From the weighted average it is unclear which category achieved which agreement and what the distribution of the various categories is. It would be enlightening, if some more statistics were shared.

Are there plans to publish the research artifacts linked to this work?

**Reasons To Accept:**

With the rise of more and more spoken material and spoken interactions dealing with peculiarities of spoken language is an important task. The paper presents a dataset, annotations and baseline experiments to deal with discfluencies, a rather prominent phenomenon in spoken language, which disturbs most NLP pipelines.

**Reasons To Reject:**

It is unclear if the data cleaned with the proposed method is actually easier/better to process for down-stream tasks, as the paper only presents an intrinsic evaluation, but not an extrinsic. As such, it is unclear to what end the disfluency removal is being carried out.

**Reproducibility:**

3: Could reproduce the results with some difficulty. The settings of parameters are underspecified or subjectively determined; the training/evaluation data are not widely available.

**Reviewer Confidence:**

4: Quite sure. I tried to check the important points carefully. It's unlikely, though conceivable, that I missed something that should affect my ratings.

---

> ### Author Rebuttal · Authors · 2023-08-27
>
> We thank the reviewer for their valuable questions and feedback. Please see our response per question/concern below.
>
> Reviewer: "It is unclear if the data cleaned with the proposed method is actually easier/better to process for down-stream tasks…"
>
> Response: Our main motivation in this work in the first place was to aid human comprehension of transcribed speech by cleaning up unnecessary parts of transcripts that do not convey semantic meaning or are redundant, making the transcript more readable by humans. While it is likely that such a clean-up technique would also benefit the downstream spoken language understanding tasks (e.g. summarization), we currently limit the scope of such explorations in this short paper and leave it for future work.
>
> Reviewer: "Is there a reason the ICSI corpus has not been used? In the past, it has been studied extensively for disfluencies. Similar to the TRAINS corpus, where also a range of scientific papers have been published on this topic."
>
> Response: Switchboard is one of the widely-used datasets for disfluency detection and has been used extensively in prior work (see for example [1,2,3] below). Switchboard consists of spontaneous multi-party speech over the phone between strangers over a diverse set of topics. While ICSI corpus also consists of multi-party conversations, the dataset is very domain-specific (a lab's group meeting recordings), thus making it a less ideal choice for the only corpora being annotated with multi-turn discontinuities. The same concerns apply to the TRAINS set. We hope that our initial annotation schema developed and used in Switchboard corpus is general enough and can be applied in future work to annotating other datasets such as the ones mentioned by the reviewer.
>
> Reviewer: "A lot of the phenomena covered in the annotations, could also be handled using Dialog Acts and more specifically, Backchannels, which some of these phenomena are."
>
> Response: We indeed considered automatic data preprocessing approaches to handle some multi-turn phenomena (see details in Appendix A.2). Although the backchannels can handle part of the “Acknowledgment” discontinuity type categorized in Table 1, we found that the remaining types cannot be resolved by Dialogue Acts, such as “Think Aloud” and Cross-turn “Repetitions” (an example in figure 1B). Therefore, we recruit humans and guide them to focus on removing these more complicated multi-turn discontinuities.
>
> Reviewer: "The paper reports the IAA based on Fleiss Kappa for the annotation. From the weighted average it is unclear which category achieved which agreement and what the distribution of the various categories is…"
>
> Response: We compute the IRR score on Fleiss Kappa in order to show that the annotation quality resulting from our schema is comparable to experts. We will add more reviewer suggested statistics given more space in the extra page upon acceptance.
>
> Reviewer: "Are there plans to publish the research artifacts linked to this work?"
>
> Response: We have released the annotated dataset described in this paper.
>
> Reviewer: "While the authors state how much the annotators (turkers) were paid, it is open, whether they were paid adequately, if they were treated fairly etc."
>
> Response: The hourly wage paid to the workers is comparable with the average wages usually used in industry for similar types of annotation work. Figure 4 presents the exact prompt presented to the anonymous workers who completed the annotation through the Amazon Mechanical Turk platform. We are not sure what the reviewer means by "if they were treated fairly" since all the infrastructure and employment relationship is handled by the Amazon Mechanical Turk platform, while the authors only provide the task (Figure 4) and the price per task ($0.40 per task + bonuses).
>
> References:
>
> [1] Honnibal, Matthew, and Mark Johnson. "Joint incremental disfluency detection and dependency parsing." Transactions of the Association for Computational Linguistics 2 (2014): 131-142.
>
> [2] Zayats, Vicky, Mari Ostendorf, and Hannaneh Hajishirzi. "Disfluency detection using a bidirectional LSTM." Interspeech (2014).
>
> [3] Dong, Q., Wang, F., Yang, Z., Chen, W., Xu, S., & Xu, B. (2019, July). Adapting translation models for transcript disfluency detection. In Proceedings of the AAAI Conference on Artificial Intelligence (Vol. 33, No. 01, pp. 6351-6358).

---

### Meta-Review · Area_Chair_5R72 · 2023-09-16

**Recommendation:** 4

**Metareview:**

The paper present multi-turn spoken conversation transcript cleaning process. They label switchborad corpus with proposed annotation shceme.

Pros:
- paper presents a dataset and baseline experiments
- labelling schema is well designed

Cons:
- it is unclear if dataset is better for downstream tasks

---

### Decision · Program_Chairs · 2023-10-07

**Decision:**

Accept-Main

**Comment:**

The paper present multi-turn spoken conversation transcript cleaning process. They label switchborad corpus with proposed annotation shceme.

Pros:
- paper presents a dataset and baseline experiments
- labelling schema is well designed

Cons:
- it is unclear if dataset is better for downstream tasks